# Venous Thromboembolism and Its Risk Factors in Children with Acute Lymphoblastic Leukemia in Israel: A Population-Based Study

**DOI:** 10.3390/cancers12102759

**Published:** 2020-09-25

**Authors:** Shlomit Barzilai-Birenboim, Ronit Nirel, Nira Arad-Cohen, Galia Avrahami, Miri Ben Harush, Assaf Arie Barg, Bella Bielorai, Ronit Elhasid, Gil Gilad, Amos Toren, Sigal Weinreb, Shai Izraeli, Sarah Elitzur

**Affiliations:** 1Department of Pediatric Hematology-Oncology, Schneider Children’s Medical Center of Israel, Petach Tikva 49202, Israel; Galia2@clalit.org.il (G.A.); gilgi@clalit.org.il (G.G.); sizraeli@gmail.com (S.I.); sarhae@clalit.org.il (S.E.); 2Sackler Faculty of Medicine, Tel Aviv University, Tel Aviv 6997801, Israel; assaf.berg@sheba.health.gov.il (A.A.B.); bella.bielorai@sheba.health.gov.il (B.B.); ronite@tasmc.health.gov.il (R.E.); amost@post.tau.ac.il (A.T.); 3Department of Statistics and Data Science, Hebrew University, Jerusalem 9190501, Israel; nirelr@mail.huji.ac.il; 4Department of Pediatric Hematology-Oncology, Ruth Rappaport Children’s Hospital, Rambam Health Care Campus, Technion-Israel Institute of Technology, Haifa 3109601, Israel; n_arad-cohen@rambam.health.gov.il; 5Department of Pediatric Hematology-Oncology, Soroka Medical Center, Ben Gurion University, Beer Sheva 84990, Israel; miribe@bgu.ac.il; 6Division of Pediatric Hematology, Oncology and Bone Marrow Transplantation, The Edmond and Lily Safra Children’s Hospital, Sheba Medical Center, Ramat Gan 52620, Israel; 7Department of Pediatric Hemato-Oncology, Sourasky Medical Center, Tel Aviv 6423906, Israel; 8Department of Pediatric Hematology-Oncology, Hadassah Hebrew University Medical Center, Jerusalem 9112102, Israel; sigalweinreb@gmail.com

**Keywords:** acute lymphoblastic leukemia, venous thromboembolism, sinus vein thrombosis, hypertriglyceridemia, thrombophilia, risk factors

## Abstract

**Simple Summary:**

Current cure rates of childhood acute lymphoblastic leukemia (ALL) surpass 90% and thus the mitigation of chemotherapy-associated toxicity has become a major goal. Venous thromboembolism (VTE) is a major toxicity of ALL therapy, but its precise prevalence, risk factors, and optimal prophylaxis are yet to be determined. This study evaluated VTE in a cohort of 1191 children, including its rate, risk factors, and long-term sequelae. VTE was found in 7.5% of the cohort, 27% severe events. Most events resolved without late sequelae. We identified four risk factors associated with VTEs: Age >10 years, high-risk ALL group, severe hypertriglyceridemia during therapy, and inherited thrombophilia. Since children with these risk factors had significantly higher rates of severe VTE, a routine evaluation may identify potential candidates for prophylactic interventions.

**Abstract:**

Venous thromboembolism (VTE) is a serious complication of acute lymphoblastic leukemia (ALL) therapy. The aim of this population-based study was to evaluate the rate, risk factors, and long-term sequelae of VTE in children treated for ALL. The cohort included 1191 children aged 1–19 years diagnosed with ALL between 2003–2018, prospectively enrolled in two consecutive protocols: ALL-IC BFM 2002 and AIEOP-BFM ALL 2009. VTEs occurred in 89 patients (7.5%). Long-term sequelae were uncommon. By univariate analysis, we identified four significant risk factors for VTEs: Severe hypertriglyceridemia (*p* = 0.005), inherited thrombophilia (*p* < 0.001), age >10 years (*p* = 0.015), and high-risk ALL group (*p* = 0.039). In addition, the incidence of VTE was significantly higher in patients enrolled in AIEOP-BFM ALL 2009 than in those enrolled in ALL-IC BFM 2002 (*p* = 0.001). Severe VTE occurred in 24 children (2%), all of whom had at least one risk factor. Elevated triglyceride levels at diagnosis did not predict hypertriglyceridemia during therapy. In a multivariate analysis of 388 children, severe hypertriglyceridemia and inherited thrombophilia were independent risk factors for VTE. Routine evaluation for these risk factors in children treated for ALL may help identify candidates for intervention.

## 1. Introduction

Despite remarkable advances in the treatment of pediatric acute lymphoblastic leukemia (ALL), toxicity is a major problem. Clinically significant venous thromboembolism (VTE) during chemotherapy has a prevalence rate of up to 15% [1] and can be associated with considerable morbidity, mortality, and therapy delays. There is limited available data regarding long-term sequelae of VTE, especially cerebral sinus venous thrombosis (CSVT). Thrombotic events are usually attributable to a combination of patient-, disease-, and treatment-related factors.

The main disease-related risk factors of VTE are increased thrombin generation during active leukemia [2,3,4,5,6,7] and the interaction of procoagulant molecules and inflammatory cytokines synthesized by the malignant cells with vascular endothelial cells [8,9].

Treatment-related thrombosis is mainly associated with the use of asparaginase. This medication compromises hepatic protein synthesis, thus reducing circulating levels of plasminogen, antithrombin, protein C, and protein S, and causing qualitative abnormalities of the von Willebrand factor [2,3,5]. Most contemporary ALL protocols have replaced native *E. coli*-asparaginase with the long-acting formula, PEG-asparaginase, which may have a different thrombogenic profile, but few studies have addressed the effect of this change on the risk of thrombosis. Corticosteroids also contribute to the hypercoagulable state by elevating levels of factor VIII, the von Willebrand factor, prothrombin, and plasminogen activator inhibitor-1 [4,10,11,12,13,14,15]. Another treatment-related risk factor for VTE is the presence of a central venous line, which accounts for more than two-thirds of thrombotic events in children in general [16,17,18].

Several patient-related risk-factors for VTE have been identified, including inherited thrombophilia, [1] high-risk ALL group, older age [19,20,21], and mediastinal mass [22].

The aims of this large, population-based study were to evaluate the rate and long-term sequelae of VTE in all children treated for ALL in Israel during the study period; to compare the occurrence of VTE between two consecutive BFM protocols using the different asparaginase formulas; and to investigate the contribution of different risk factors of VTE in this patient group. Importantly, the study cohort included all children with ALL who were prospectively enrolled in national treatment protocols, thus representing an unselected population.

## 2. Results

### 2.1. Patient Characteristics

The study cohort consisted of 1191 patients whose characteristics are described in Table 1.

There were no statistically significant differences in these characteristics between children from different medical centers (Appendix A).

### 2.2. Prevalence and Characteristics of VTE Events

VTEs were documented in 89 children (7.5%) during ALL therapy; 24 (27%) were severe (the Ponte di Legno Toxicity Working Group; PTWG grades 3–4, Appendix A). Most of the events occurred during induction (70.6%) or delayed intensification (18.5%). The median interval between the last dose of asparaginase and the VTE event was 13.3 days (range 1–46) for native *E. coli*-asparaginase and 20 days (range 1–135) for PEG-asparaginase. The median interval between the last dose of corticosteroids and the VTE was 9 days (range 0–75).

### 2.3. Sites of VTE Events

Deep vein thromboses (DVT) of the limbs or right atrial thromboses (usually considered central catheter-related), accounted for 73% of all VTEs, and pulmonary embolism (PE) accounted for 3%. Cerebral sinus VTE (CSVT) occurred in 21 patients (24% of all VTEs, 1.8% of the whole cohort, Figure 1).

Neurological symptoms of cerebral sinus vein thrombosis (CVST) included convulsions (8), hemiparesis (6), severe headaches (3), and coma (2). Three patients (all younger than 10 years of age) presented with milder symptoms: Moderate headache (2), earache (1), and irritability (1). In two children, cerebral sinus venous thrombosis (CSVT) recurred after re-exposure to PEG-asparaginase, despite therapeutic low-molecular-weight heparin (LMWH) administration.

There was no correlation between central nervous system leukemic involvement at the diagnosis of ALL and the presence of CSVT (*p* = 0.21) and no difference in mortality or relapse rates between children with and without CSVT (Appendix A).

### 2.4. Risk Factors for VTE; Univariate Analysis

Analysis by risk factors (Table 2) yielded a significantly higher rate of VTE in children older than 10 years than in younger children (*p* = 0.015). The average age of children without VTE was 7.2 years (range 1.1–19.3), children with mild VTE: 8.3 years (range 1.2–19.1), and those with severe VTE: 10.3 years (range 1.7–18.5) (*p* < 0.001). VTEs were also significantly more common in children in the high-risk ALL group compared to the standard or medium-risk groups (11.2% vs. 6.4%, *p* = 0.039) and in children treated with the AIEOP-BFM ALL 2009 protocol (including PEG-asparaginase) than in children treated with the ALL-IC BFM 2002 protocol (including native *E. coli*-asparaginase) (10.2% vs. 4.7% *p* = 0.001 for any VTE; and 3.0% vs. 1.0%, *p* = 0.010 for severe VTE). In addition, VTE rates were significantly associated with severe hypertriglyceridemia (*p* = 0.005).

Thrombophilia screening was performed in 584 children attending Schneider Children’s Medical Center of Israel (SCMCI) and Ruth Rappaport Children’s Hospital (RR). Findings were positive in 84 (14.4%). Children with thrombophilia had significantly more VTEs than children without thrombophilia (*p* < 0.001).

Sex, body mass index (BMI) percentile, and leukemic cell lineage did not significantly affect the occurrence of VTE.

The risk factors for VTE found in our study were cumulative. All 24 children with severe VTE (PE and CSVT) had at least one risk factor, and 13 had more than one (Figure 2, Table 3).

#### Hypertriglyceridemia in Children with ALL

Abnormal triglyceride levels were found in the majority of those who were tested (*n* = 584), at least at one time-point: Grade 1 hypertriglyceridemia (>150 mg/dL) was found in 91.3%, grade 3 (>500 mg/dL) in 32.5% (*n* = 190), and grade 4 (>1000 mg/dL) in 11.5% (*n* = 67). Severe hypertriglyceridemia (grade ≥3) was not associated with BMI percentile (adjusted for age), (Appendix A).

Importantly, hypertriglyceridemia (≥500 mg/dL) was significantly more common in children with VTE and was found in 32.5%, 54.8%, and 61.1% of the children without, with mild, and with severe VTE, respectively (*p* = 0.001) (Figure 3, Appendix A).

Analysis by event severity revealed a median triglyceride level of 711 (121–7190 mg/dL) in children with mild VTE, and 1118 (118–4163 mg/dL) in children with severe VTE, compared to 542 (55–5510 mg/dL) in children without VTE. Furthermore, 14.1% of the children with hypertriglyceridemia grade 3 (500–1000 mg/dL) and 16.4% of those with hypertriglyceridemia grade 4 (>1000 mg/dL) developed VTE, compared to 5.3% of VTE among children with triglycerides levels <500 mg/dL (*p* = 0.001). This association was even more prominent for severe VTE, with 5.8% and 10.4% of severe VTE among children with hypertriglyceridemia grades 3 and 4 respectively, compared to 1.8% of severe VTE in children with triglyceride levels <500 mg/dL (*p* = 0.006), suggesting an association between the severity of hypertriglyceridemia and the risk of VTE.

Severe hypertriglyceridemia appeared mainly during two-time points: After 4–6 weeks of therapy, or after 5–7 months of therapy, namely after the treatment phases in which asparaginase was administered and was not found to be associated with steroid type or with older age (Figure 4 and Appendix A).

Each dot represents the timepoint from diagnosis and the value of maximal triglycerides level for a single patient. The horizontal line indicates a triglyceride level of 500 mg/dL; Abbreviations: TG, triglyceride; Max, maximal value; VTE, venous thromboembolism.

Hypertriglyceridemia was more common in children treated according to the AIEOP-BFM ALL 2009 protocol (PEG-asparaginase) than in those treated according to ALL-IC BFM 2002 (native *E. coli*-asparaginase) (*p* = 0.013; Appendix A). Interestingly, among 14.4% of the tested patients, the maximal triglyceride level was measured at ALL diagnosis, before therapy initiation (Figure 4). Triglyceride levels at diagnosis were not predictive of future severe hypertriglyceridemia during therapy. None of the patients had grade 4 hypertriglyceridemia at ALL diagnosis. Only eight children (1.4%) had grade 3 hypertriglyceridemia at diagnosis, and none developed VTE.

### 2.5. Risk Factors for VTE; Multivariate Analysis

A multivariate analysis of all risk factors for VTE, namely age, gender, treatment protocol, thrombophilia, ALL phenotype, risk group, BMI percentile, and hypertriglyceridemia was performed for all children from SCMCI (388) who were uniformly evaluated for all 8 risk factors. Thrombophilia and hypertriglyceridemia ≥ grade 3 were found to be independently predictive of VTE (Table 4).

### 2.6. Outcome of VTE

Long-term sequelae of VTEs included one case of lower limb post-thrombotic syndrome in a patient with DVT and residual convulsive disease in two patients with CSVT. There were no other major long-term neurologic effects in patients with CSVT. Significant bleeding events did not occur in patients treated with LMWH.

In 17 patients with VTE (10 with CSVT), the event resulted in a modification of the treatment protocol consisting of cessation or reduction of asparaginase or methotrexate therapy (Table 3).

There were no significant differences in the rate of complete remission, death, or relapse between children with or without VTE (Appendix A).

## 3. Discussion

In this population-based study, including all pediatric patients diagnosed in Israel with ALL and enrolled in two prospective BFM-based therapeutic protocols between the years 2003–2018, the rate of VTE was 7.5%. CSVT accounted for 24% of all VTEs. As noted in earlier studies, most events occurred during the induction or delayed intensification phase of therapy, in temporal proximity to asparaginase and corticosteroid therapy [1,16]. The most common presenting symptoms of CSVT were hemiparesis, seizures, or severe headaches that led to urgent diagnostic brain imaging. Several younger children presented with non-specific, milder symptoms such as moderate headache, earache, or irritability. We suggest that a higher level of suspicion for CSVT may be appropriate within this younger age group.

Over the 15-year study period, the prevalence of major long-term sequelae of VTEs was low, including only one event of post-thrombotic syndrome and two patients with residual convulsive disease post-CSVT. Long-term follow-up with directed neurological evaluation is required to rule out late and milder neurologic sequelae of CSVT.

VTE led to treatment modifications in 17 children (18.5%), of whom 10 had CSVT, but unlike previous studies [23,24], rates of death, relapse, and event-free survival were similar in children with and without VTE. This may be due to early diagnosis of CSVT events, and prompt intervention, leading to minor treatment modifications only.

In univariate analysis, we identified four risk factors for VTE in our cohort: Age ≥ 10 years (*p* = 0.015), high-risk ALL group (*p* = 0.039), inherited thrombophilia (*p* < 0.001), and hypertriglyceridemia grade 3–4 (*p* = 0.005). In addition, VTEs were more common when therapy included PEG-asparaginase as opposed to native *E. coli*-asparaginase (*p* = 0.001). The first three factors have been reported previously [1,19,20,21,22,25], but until recently, hypertriglyceridemia has been mentioned only seldomly in association with VTE in children with ALL [26,27]. Furthermore, hypertriglyceridemia was also indicated by us as an independent risk factor for VTE in the multivariate analysis.

As confirmed by our study, hypertriglyceridemia is common in children with ALL. Cohen et al. [28] reported a mean triglyceride level of 459 ± 526 mg/dL in children with ALL during treatment. Both asparaginase and corticosteroid treatment may contribute to hypertriglyceridemia. Asparaginase impairs hepatic mitochondrial function, leading to increased levels of cholesterol, phospholipids, and triglycerides, and to alterations in very low-density lipoprotein metabolism and severe fatty metamorphosis [29,30]. Corticosteroids are adipokinetic agents, causing lipolysis and triglyceride breakdown. Corticosteroids are also involved in the mobilization and redistribution of body fat and increase the synthesis of triglyceride and the activity of lipoprotein lipase, a key enzyme in triglyceride hydrolysis [26,31]. However, the presence of severe hypertriglyceridemia at ALL diagnosis may suggest additional pathogenic mechanisms contributing to hypertriglyceridemia besides those associated with corticosteroid and asparaginase exposure.

Grades 3 or 4 hypertriglyceridemia, at any time point, were found in 32.5% (*n* = 190) of the tested children, of whom 5.8% had severe VTE compared to 1.8% of severe VTE in children with triglyceride levels < 500 mg/dL.

Bhojwani et al. [26] found severe hypertriglyceridemia during chemotherapy in 7% of 257 children with ALL, and this subgroup had a two- to three-fold higher risk of thrombosis than patients without hypertriglyceridemia. These findings were recently supported by another study [27] reporting hypertriglyceridemia in 10.5% of the standard/high risk ALL with a significant association with thrombosis.

In our study, both VTE and hypertriglyceridemia were more common in children treated according to the AIEOP-BFM ALL 2009 protocol (PEG-asparaginase) than in children treated according to the ALL-IC BFM 2002 protocol (native *E. coli*-asparaginase). Since all patients randomized to additional doses of PEG-asparaginase within the AIEOP-BFM 2009 protocol were excluded from the study, our results may imply a difference in adverse event rates between the two preparations, a finding that was not supported by previous studies [10,32] until recently [27]. It is possible that due to its longer half-life, the pegylated formula has a more extended metabolic and hepatotoxic effect, thereby increasing the risks of both hypertriglyceridemia and VTE.

The relationship between VTE and hypertriglyceridemia is not completely clear. Does hypertriglyceridemia serve only as a surrogate marker for the stronger pro-thrombotic effect of asparaginase, or does it really contribute to the increased risk of VTE? Metabolic syndrome, including hyperlipidemia, has been suggested as a risk factor for VTE [33,34,35]. However, most studies have failed to demonstrate such a relationship [36]. Hypertriglyceridemia has been associated with increased thrombin generation, elevated factor VII, VIII, IX, fibrinogen, and plasminogen activator inhibitor, and increased blood viscosity [35]. This question may have important therapeutic implications and should be further investigated.

Another interesting question is whether hypertriglyceridemia at diagnosis can predict an elevated risk of VTE. In our study, neither hypertriglyceridemia at ALL diagnosis nor BMI percentile were predictive of hypertriglyceridemia during therapy. It seems that as in VTE, the pathogenesis of hypertriglyceridemia is multifactorial and derives from a combination of patient, disease, and therapy factors. The significant number of children who demonstrated elevated baseline levels of triglycerides may imply a genetic predisposition contributing to hypertriglyceridemia. Future studies, including genome-wide association analyses, may identify children with an elevated risk for severe hypertriglyceridemia, for whom pharmacological intervention may be warranted.

Prophylactic LMWH therapy in children with inherited thrombophilia has been shown to be safe and effective in reducing the incidence of VTE [25,37,38,39]. In light of our current findings, it may be reasonable to consider therapeutic interventions for other groups at high-risk for thrombosis as well. Several thrombophylactic modalities may be employed during the therapy of childhood ALL. One option is LMWH, as described above. However, the full anticoagulative effect of LMWH requires the presence of antithrombin, yet during ALL therapy there is a decrease in antithrombin levels as a result of the hepatotoxic effect of asparaginase. Accordingly, antithrombin substitution, with or without LMWH, was found to be effective in reducing VTE during treatment with asparaginase [40].

The THROMBOTEC study [41] has recently demonstrated that antithrombin and LMWH are equally effective in preventing thromboembolic events in children with ALL. The authors recommended LMWH as the primary choice due to an increased leukemia relapse rate within the antithrombin arm; however, one-third of the cohort were non-compliant with LMWH thromboprophylaxis due to the difficulties of subcutaneous injection administration in the pediatric population.

The new oral anticoagulants (NOAC) directly inhibit thrombin or factor Xa and do not require antithrombin for their activity. In our study, a second CSVT developed in two children after re-exposure to PEG-asparaginase while receiving therapeutic LMWH. Future use of NOACs, whose mechanism may be more suitable to treat VTE in patients with ALL, may potentially contribute to the prevention of such events. Further studies are needed once these drugs are approved for the pediatric population. Meanwhile, we recommend that a carefully selected high-risk group should be offered thromboprophylaxis with LMWH.

The limitation of this study is mainly that triglyceride levels and thrombophilia were fully evaluated in half of the cohort (584 children from 2 tertiary hospitals: SCMCI and RR). The logistic regression analysis was performed only on children from SCMCI (*n* = 388) because children with thrombophilia from RR received LMWH prophylaxis. However, this subgroup did not significantly differ from the rest of the cohort (Appendix A).

## 4. Conclusions

In this population-based study of 1191 children with ALL, VTEs were found in 7.5%, while CSVT accounted for 24% of all events. Most VTE resolved without late sequelae, and morbidity and mortality rates were low. We identified four potential risk factors for VTEs; in addition to age >10 years, high-risk ALL group, and inherited thrombophilia, which were previously identified as risk factors, this study highlights an association with severe hypertriglyceridemia during therapy. In addition, VTEs were significantly more common during treatment according to the AIEOP-BFM 2009 protocol (PEG-asparaginase) in comparison to the ALL-IC BFM 2002 protocol (*E. coli*-asparaginase). Since children with these risk factors had significantly higher rates of severe VTE, routine evaluation and thromboprophylaxis might be considered. Future studies revealing the precise pathogenesis and relationship between VTE and hypertriglyceridemia may aid in identifying appropriate candidates for therapeutic interventions

## 5. Methods

### 5.1. Patients and Setting

The study cohort included all 1191 children aged 1–19 years newly diagnosed with ALL between 2003 and 2018 at any of the seven pediatric hematology–oncology centers in Israel. All patients were prospectively enrolled in two treatment protocols: 591 children were treated from 2003 to 2010 according to the ALL-IC BFM 2002 (NCT 00764907) protocol, and 600 children were treated from 2011 to 2018 according to the AIEOP-BFM ALL 2009 (NCT 1117441) protocol. Thirty-six children who were treated according to the AIEOP-BFM ALL 2009 protocol and randomized to receive additional doses of PEG-asparaginase were excluded from our study cohort (Figure 5).

In two major medical centers, Schneider Children’s Medical Center of Israel (SCMCI) and Ruth Rappaport Children’ Hospital (RR), all 584 children diagnosed with ALL underwent a comprehensive evaluation, including screening for Factor V Leiden and prothrombin G20210A mutations. Children from SCMCI (*n* = 388) were also evaluated for APLA, LP(a), and homocysteine levels [37]. In addition, this group underwent monitoring of triglyceride levels during therapy, and echocardiographic evaluation (induction and delayed intensification) (Figure 5).

Patients diagnosed with VTEs were treated with a therapeutic dose of low-molecular-weight heparin (LMWH), 1 mg/kg, twice daily. During heparin treatment, a minimum platelet threshold of 50,000 per microliter was maintained. Prophylactic LMWH (1 mg/kg/day) was administered to all 25 children with congenital thrombophilia at RR hospital, in accordance with institutional policy, as described elsewhere. [37]

### 5.2. Data Collection

The study was approved by the ethical committees of all participating hospitals. Informed consent was signed by parents or legal guardians. VTEs were prospectively captured as adverse events, according to protocol definitions, including deep venous thrombosis, CSVT, and any other grade 4 thrombosis (e.g., pulmonary embolism, arterial thrombosis).

In addition, medical records were retrospectively reviewed for VTE details, patient and disease characteristics, laboratory data including triglyceride level and coagulation tests, presence of inherited thrombophilia, radiological findings, and course of chemotherapy. The entire cohort (*n* = 1191) underwent a full evaluation for five potential risk factors including age, gender, ALL risk group, ALL lineage, and treatment protocol. In 584 children from two tertiary medical centers (RR and SCMCI), two additional potential risk factors were evaluated: Thrombophilia and hypertriglyceridemia. Detailed data regarding thrombophilia screening and prophylaxis were also described in our previous report [37]. VTEs and triglyceride levels were graded by severity according to the classifications of the Common Terminology Criteria for Adverse Events (CTCAE), version 4.03, and the Ponte di Legno Toxicity Working Group (PTWG); Severe VTE and severe hypertriglyceridemia were defined as grades 3–4 [42,43] (Appendix A).

### 5.3. Statistical Analysis

We used the phi coefficient of correlation to estimate associations in 2 × 2 tables [44] and the Pearson coefficient to measure the association between continuous variables. A Fisher’s exact test was used to compare frequency distributions of categorical variables between groups, and the Wilcoxon-Mann–Whitney test was used to compare continuous data. A two-sided *p*-value of 0.05 or less was considered statistically significant. Data regarding 388 children from one tertiary medical center (SCMCI), were entered into a logistic regression model to evaluate the association of thrombophilia (yes vs. no) and hypertriglyceridemia grade (1–2 vs. 3–4) with VTE after controlling for sex, age at diagnosis (≥10 years vs. <10 years), ALL protocol (ALL-IC BFM 2002 vs. AIEOP-BFM 2009), lineage (T vs. non-T), BMI percentile, and ALL risk group (high-risk vs. non-high-risk). The multivariant analysis was performed for this group since children treated in SCMCI were uniformly assessed for all variants and did not receive thromboprophylaxis.

Results are reported as odds ratios and 95% confidence intervals. The predictive performance of the model was calculated using the area under the receiver operating characteristic curve (AUC) [45] All statistical analyses were performed with the R Project for Statistical Computing, version 3.4.2.

## Figures and Tables

**Figure 1 cancers-12-02759-f001:**
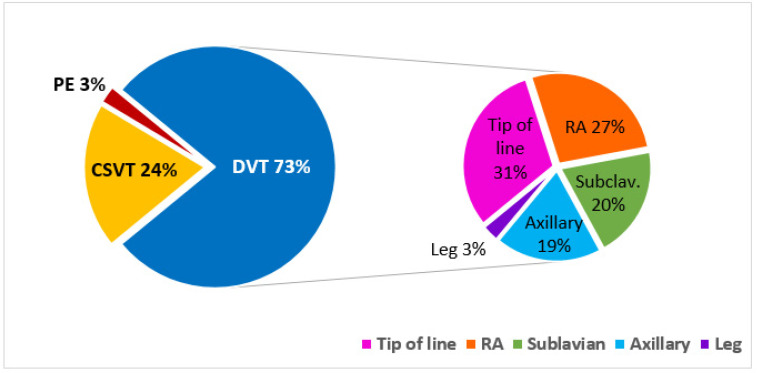
Type and site of venous thromboembolism (VTE). Abbreviations: CSVT, cerebral sinus vein thrombosis; PE, pulmonary emboli; DVT, deep vein thrombosis; RA, right atrium; Axillary, axillary vein; Subclav., subclavian vein.

**Figure 2 cancers-12-02759-f002:**
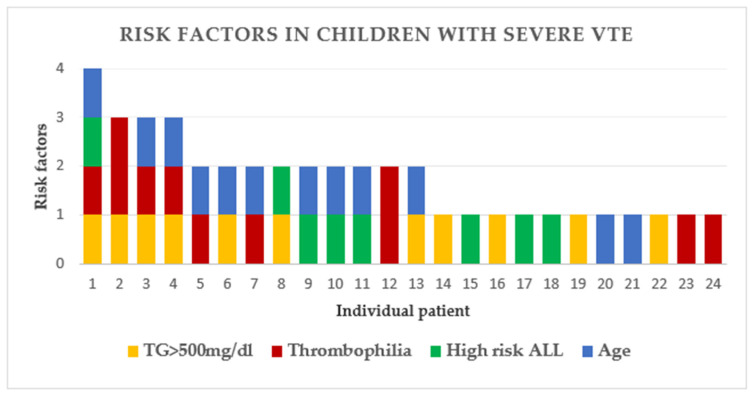
Cumulation of 4 risk factors for severe VTE—univariate analysis Each vertical line represents a single patient. Each risk factor is represented by a different color. Hypertriglyceridemia and thrombophilia screening were performed in 584 children attending SCMCI and RR; Patients 2 and 12 had two mutations of thrombophilia (see also Table 3); Abbreviations: TG, triglyceride; HR ALL, high risk ALL; VTE, venous thromboembolism.

**Figure 3 cancers-12-02759-f003:**
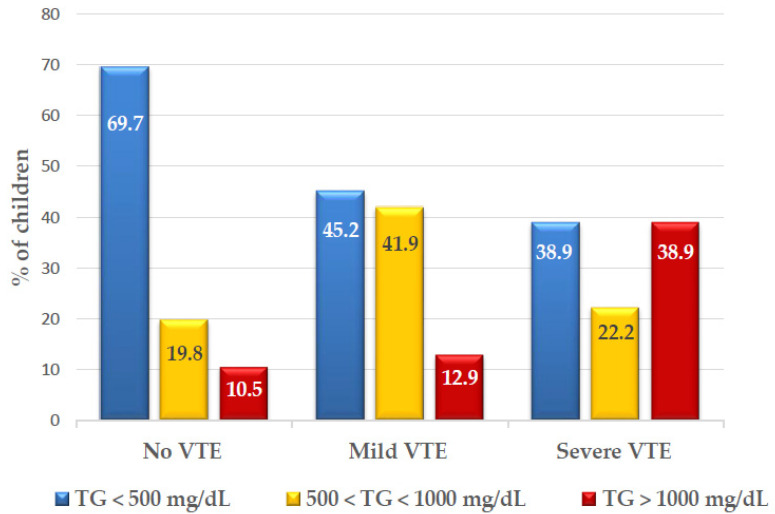
Triglycerides levels according to VTE groups: (*n* = 584). Abbreviations: TG, triglyceride VTE, venous thromboembolism.

**Figure 4 cancers-12-02759-f004:**
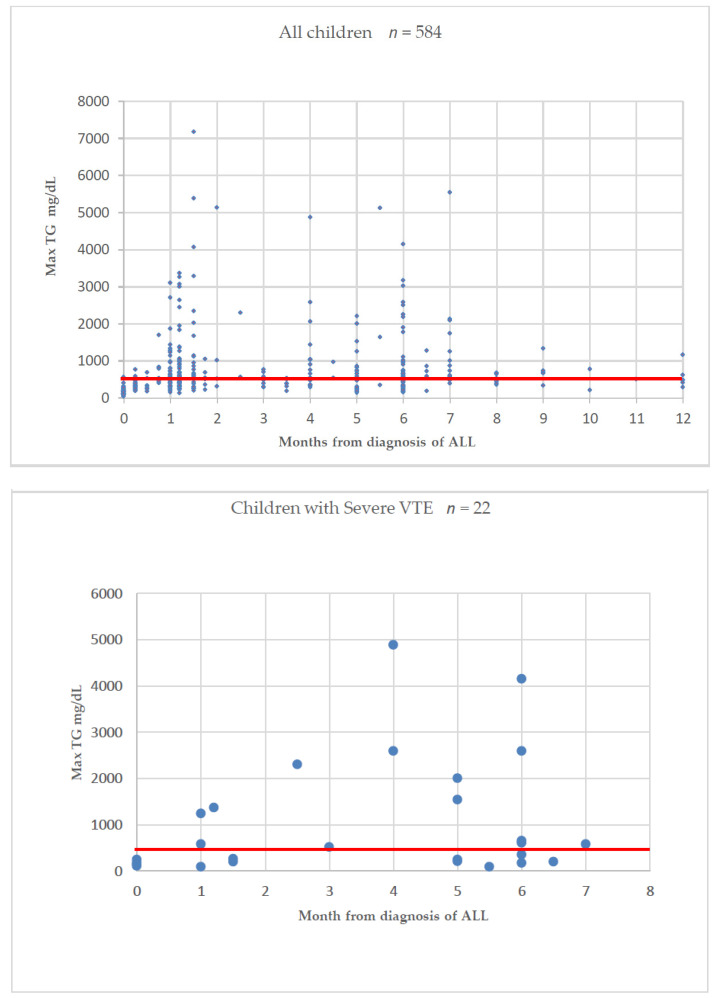
Time of maximal triglycerides level among all tested children and among children with severe VTE.

**Figure 5 cancers-12-02759-f005:**
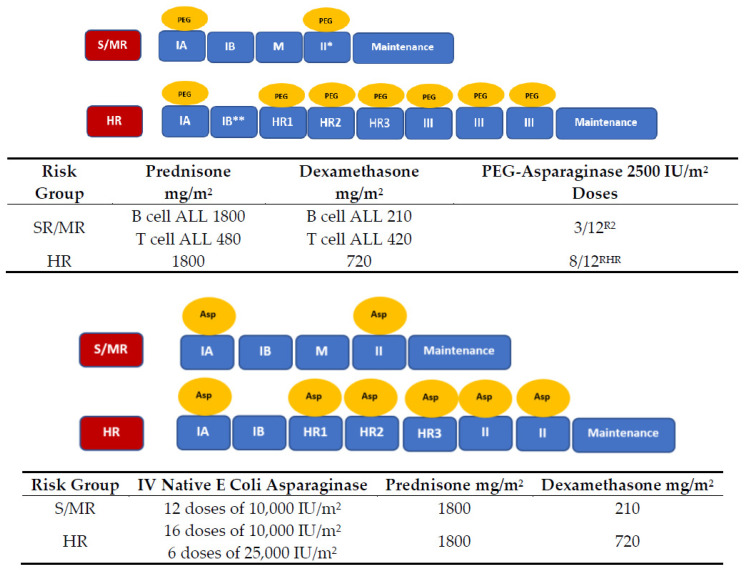
ALLIC- BFM 2002 and AIEOP- BFM ALL 2009 protocols. TG measurements: Twice a week during asparaginase therapy. Echocardiogram: Before blocks IA and IIA; AIEOP-BFM ALL 2009 protocol: * MR randomized (R_2_) to receive extra 9 doses of PEG; ** HR randomized (R_HR_) to receive extra 4 doses of PEG-asparaginase; all children randomized to extra doses of PEG (R_2_ and R_HR_) were excluded from the study. Abbreviations: HR, high risk; S/MR, standard/medium risk; PEG, PEG-asparaginase; Asp, *E.-Coli*-asparaginase, VCR; Vincristine, DNR; Daunorubicin, CPM; Cyclophosphamide, ARA-C; Cytarabine, 6MP; 6-Mercaptopurine, HDMTX; High-dose Methotrexate, DOXO; Doxorubicin, 6TG; 6-Thioguanine, HD-ARA-C; High-dose Cytarabine, IFO; Ifosfamide, VP16; MTX, Methotrexate; R2, rRandomization for the MR group; RHR, Randomization for the HR group.

**Table 1 cancers-12-02759-t001:** Characteristics of 1191 patients with acute lymphoblastic leukemia (ALL).

Characteristic	Value
Sex	
Female	501 (42.1%)
Male	690 (57.9%)
Age (year), median (range)	6 (1.02–19.30)
ALL treatment risk group	
High risk	262 (22.0%)
Non-high risk	929 (78.0%)
Lineage	
T	190 (16.0%)
B	1001 (84.0%)
Treatment protocol *	
2002	591 (49.6%)
2009	600 (50.4%)

* ALL-IC-BFM 2002; AIEOP BFM ALL 2009.

**Table 2 cancers-12-02759-t002:** Risk factors for VTE in children with ALL—univariate analysis.

Risk Factors	Children *n* (%)	No VTE	VTE	*p*-Value * VTE vs. No VTE	Severe VTE	*p*-Value ^†^ Severe VTE vs. No VTE	Mild VTE	*p*-Value ^†^ Mild VTE vs. No VTE
All Cohort	*n* = 1191	*n* = 1102	*n* = 89		*n* = 24		*n* = 65	
Protocol								
BFM 2002 BFM 2009	591 (49.6%) 600 (50.4%)	563 539	28 (4.7%) 61 (10.2%)	0.001	6 (1.0%) 18 (3.0%)	0.010	22 (3.7%) 43 (7.2%)	0.007
Age group								
<10 years ≥10 years	840 (70.5%) 351 (29.5%)	789 313	51 (57.3%) 38 (42.7%)	0.015	12 (50%) 12 (50%)	0.037	39 (60%) 26 (40%)	0.050
Risk group								
HR Non-HR	262 (22.0%) 929 (78.0%)	233 869	29 (32.6%) 60 (67.4%)	0.039	8 (33.3%) 16 (66.7%)	0.202	21 (32.3%) 44 (67.7%)	0.042
Lineage								
T B	190 (16.0%) 100 (84.0%)	175 927	15 (16.8%) 74 (83.2%)	0.921	4 (16.7%) 20 (83.3%)		11 (16.9%) 54 (83.1%)	
Sex								
Female Male	501 (42.1%) 690 (57.9%)	470 632	31 (34.8%) 58 (65.2%)	0.179	6 (25%) 18 (75%)		25 (38.5%) 40 (61.5%)	
SCMCI ^γ^	*n* = 388		*n* = 31		*n* = 15		*n* = 16	
TG								
≤500 mg/dL ^ψ^ > 500 mg/dL	285 (73.5%) 103 (26.5%)	270 87	15 (48.4%) 16 (51.6%)	0.005	7 (46.7%) 8 (53.3%)	0.029	8 (50%) 8 (50%)	0.035
Thrombophilia								
Yes No	52 (13.4%) 336 (86.6%)	37 320	15 (48.4%) 16 (51.6%)	<0.001	9 (60%) 6 (40%)	<0.001	6 (37.5%) 10 (62.5%)	0.005
BMI%		46.7 ± 32.4	45.1 ± 33.6	0.613	50.7 ± 32.8		39.5 ± 34.6	

* Fisher’s exact test for VTE; ^†^ Pairwise tests; ^ψ^ Common Terminology Criteria for Adverse Events (CTCAE) grades 3–4, (Appendix A); ^γ^ Thrombophilia and hypertriglyceridemia were analyzed as risk factors for VTE only in SCMCI; Severe VTE was defined as grades 3–4 by the PTWG classification (Appendix A). Abbreviations: HR, high risk; nHR, non-high risk; BFM 2002, ALL-IC BFM 2002; BFM 2009, AIEOP-BFM 2009; TG, triglycerides; SCMCI, Schneider Children’s Medical Center of Israel.

**Table 3 cancers-12-02759-t003:** Characteristics of 24 patients with severe VTE events ^©.^

Pt.	Age (year)	CL	Phase	Event	TG Diag	Max TG	Time Max TG	RF +	Symptoms	Risk Group	Thrombophilia	ASP. Formula	Modifications	Status	Relapse
1	12.1	Port	IA	CSVT	275	526	3 m	4	Coma	B-HR	FII	PEG	Stop PEG	Alive	yes
2	5.4	Port	HDMTX	CSVT	259	1554	5 m	3	Mild headaches	B-S/MR	FVL + FVL	PEG	No	Alive	No
3	13.9	Port	IA	CSVT	261	2600	6 m	3	Hemiparesis & dysarthria	T-nHR	Lp(a)	PEG	Stop PEG	Alive	No
4	18.5	Port	IB	CSVT	179	662	6 m	3	Coma	T-nHR	Homocy	PEG	Stop PEG	Alive	No
5	13.7	Port	IB	CSVT	113	267	6 w	2	Hemiparesis & dysarthria	B-S/MR	APLA	E coli	No	Alive	No
6	13	Pick	IA	CSVT	210	4163	6 m	2	SZ & hemiparesis	B-S/MR	no	E coli	Stop ASP	Alive	No
7	15.4	Port	IB	CSVT	61	360	6 m	2	Severe headaches	B-S/MR	FVL	PEG	No	Alive	No
8	9	Port	IB	CSVT	204	2309	5 m	2	Earaches	B-HR	no	PEG	No	Alive	No
9	15	Port	IIB	CSVT	117	148	3 w	2	Hemiparesis & facialis	B-HR	no	E coli	No	Alive	No
10	15.5	Port	IIIA	PE	NA	208	NA	2	Chest pain & dyspnea	B-HR	no	E coli	No	Alive	No
11	16	Pick	IA	CSVT	NA	NA	NA	2	SZ	B-HR	NA	E coli	Stop PEG	Alive	No
12	4.1	Port	IB	CSVT	127	127	1	2	Mild headaches & fatigue	B-S/MR	FII + FVL	PEG	No	Alive	No
13	10.9	Port	IB	CSVT	113	610	6 m	2	SZ	B-S/MR	no	PEG	Stop PEG	Alive	No
14	1.7	Port	IA	CSVT	87	1247	4 w	1	Severe headaches & earaches	B-S/MR	no	PEG	No	Alive	No
15	6.8	Port	IA	PE	72	250	4 w	1	Tachycardia	T-HR	no	PEG	No	Alive	No
16	7	Port	IB	CSVT	168	1370	5 w	1	SZ	B-S/MR	no	PEG	No	Alive	No
17	7	Pick	HR3	CSVT	NA	211	NA	1	SZ	B-HR	no	PEG	No	Dead	Yes
18	7.2	Port	IA	CSVT	NA	100	NA	1	SZ	B-HR	no	PEG	Stop PEG	Alive	No
19	9.2	Port	IA	CSVT	228	2013	5 m	1	Status epilepticus	B-S/MR	no	PEG	Postpone IT	Alive	yes
20	13.2	Pick	IIB	CSVT	NA	NA	NA	1	Hemiparesis	T-nHR	NA	E coli	No	Alive	No
21	17.6	Port	IA	CSVT	NA	98	NA	1	SZ and syncope	B-S/MR	NA	PEG	Postpone IB	Alive	No
22	4.9	Port	IB	CSVT	256	582	4 w	1	Severer headaches	B-S/MR	no	PEG	No	Alive	No
23	7.2	Port	IIA	PE	95	175	6 m	1	Tachycardia & tachypnea	B-S/MR	Lp(a)	PEG	No	Alive	No
24	2.4	Port	IB	CSVT	108	206	5 m	1	SZ	B-S/MR	FII	PEG	Stop PEG	Alive	No

**^©^** Out of 89 children with VTEs among 1191 patients with ALL; ^+^ Four risk factors for VTE: Triglycerides >500 mg/dL, thrombophilia, age >10 years, high-risk ALL group; Abbreviations: RF, risk factors; CL, central line; CSVT, cerebral sinus vein thrombosis; PE, pulmonary emboli; Pick, pick line; SZ, seizures; VTE, venous thromboembolism; Homocy, elevated homocysteine; APLA, anti-phospholipid antibody; FVL, factor V Leiden; FII, prothrombin G20210A mutation; Lp(a), lipoprotein A; NA, not available; nHR, non-high risk; S/MR, standard/medium risk; HR, high risk; TG diag, triglycerides at diagnosis of ALL; Max.TG, maximum triglycerides level; VTE, venous thromboembolism.

**Table 4 cancers-12-02759-t004:** Logistic regression of risk factors for VTE done for 388 children *.

Covariate	Reference	OR	95% CI	*p*-Value
Age	<10 years	0.95	0.36, 2.35	0.91
Gender	Female	1.46	0.61, 3.74	0.41
BMI percentile		0.99	0.98, 1.01	0.30
Protocol	ALL-IC BFM 2002	2.30	0.97, 5.83	0.07
Thrombophilia	No	10.58	4.41, 26.28	<0.001
Lineage	Non-T	0.80	0.23, 2.40	0.71
Risk group	Non-HR	0.63	0.20, 1.75	0.40
Severe hyper-TG ≥ 500 mg/dL	Non-severe	3.90	1.59, 9.78	0.003

AUC: 0.78; McFadden Pseudo R2: 0.23; * Multivariate analysis included all children from SCMCI (uniformly captured for all 8 variables); SCMCI, Schneider Children’s Medical Center of Israel; BMI, body mass index; HR, high risk; TG, triglycerides; AUC, area under the receiver operating characteristic curve.

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
