# Peer review of "Venous Thromboembolism and Its Risk Factors in Children with Acute Lymphoblastic Leukemia in Israel: A Population-Based Study"

_cancers, 2020, doi:10.3390/cancers12102759_

Round 1

Reviewer 1 Report

The authors present a large prospective population-based study with the aim to evaluate the rate, risk factors and long-term sequelae of VTE in children with ALL. Four risk factors identified to be associated with the VTE risk: older age, high risk ALL, severe hypertriglyceridemia and inherited thrombophilia.

The manuscript provides important information to pediatric oncologists and adds knowledge on the risk factors for thrombosis in pediatric ALL patients. The article is well written, well organized and easy to read. The limitation is, however, that triglyceride levels and thrombophilia tests were investigated in half of the patients only. Furthermore, only half of the latter were tested for thrombophilia.

Questions/Comments

  1. The authors found higher incidence of hypertriglyceridemia in children treated according to the protocol with PEG-asparaginase than in those treated according to the protocol with native E. coli-asparaginase providing with the insight that different formulations may have a different thrombogenic profile. This goes in line with findings from other groups (Finch ER, 2020). Some studies reported asparaginase activity levels in patients receiving PEG-asparaginase to be consistently higher compared to other asparaginase formulations (Tong WH, 2014). Was asparaginase levels measured for the patients included into the study on both protocols?
  2. As dyslipidaemia is thought to be a result of an effect of combined treatment with asparaginase and corticosteroids: did the authors include steroids type (Pred vs Dexa) into analysis as potential risk factor?
  3. Did the authors compare the levels of hypertriglyceridemia between two protocols?

 Minor comment

Figure 1. Pie diagram reflects nicely the findings, and the table below should be deleted as it duplicates the data.

Author Response

Comment 1: The authors found higher incidence of hypertriglyceridemia in children treated according to the protocol with PEG-asparaginase than in those treated according to the protocol with native E. coli-asparaginase providing with the insight that different formulations may have a different thrombogenic profile. This goes in line with findings from other groups (Finch ER, 2020). Some studies reported asparaginase activity levels in patients receiving PEG-asparaginase to be consistently higher compared to other asparaginase formulations (Tong WH, 2014). Was asparaginase levels measured for the patients included into the study on both protocols?

Response:  

Thank you very much for your comments. Routine measurement of PEG-asparaginase activity levels was performed only for patients enrolled in the AIEOP-BFM ALL 2009 trial, as part of the international study. Activity levels were not measured for patients receiving native E. coli asparaginase according to the ALL IC BFM 2002 trial.

Comment 2:  As dyslipidemia is thought to be a result of an effect of combined treatment with asparaginase and corticosteroids: did the authors include steroids type (Pred vs Dexa) into the analysis as a potential risk factor?

Response:

We indeed investigated this issue as you have smartly commented on. Steroid type was not found to be a risk factor for hypertriglyceridemia in our study. All children treated according to the BFM protocols receive prednisone (60mg/m2 /30d) during the induction period and then, dexamethasone (20mg/m2 /21d) during the delayed intensification period. In the AIEOP- BFM ALL 2009 protocol children with T cell ALL and GPR at D8 of induction, received dexamethasone for 21 days instead of prednisone. Severe hypertriglyceridemia (>500 mg%) was found in 191 children throughout the therapy period during both prednisone and dexamethasone treatment (table 4).  Severe hypertriglyceridemia was found in 88 children while receiving dexamethasone therapy, in 85 children while receiving prednisone therapy, and in 18 children who did not receive steroids during the preceding month (p=ns).  According to your comment we add these finding to our manuscript (line-176) 

Comment 3:  Did the authors compare the levels of hypertriglyceridemia between two protocols?

Response:  

We compared the levels of hypertriglyceridemia between the two protocols and found that hypertriglyceridemia was much more common in children treated during the AIEOP- BFM ALL 2009 protocol (Peg-asparaginase) compared to children treated during the ALL-IC BFM 2002 protocol ( E-coli- asparaginase) (p= 0.013, Table S5).

 Minor comment:

Figure 1. Pie diagram reflects nicely the findings, and the table below should be deleted as it duplicates the data

Response:  Corrected (lines 105-106).

Reviewer 2 Report

I have reviewed the article by Barzilai-Birenboim and colleagues “Venous thromboembolism and its risk factors in children with acute lymphoblastic leukemia in Israel: a population-based study”. In this article, the authors studied the risk factors to develop VTE in relation to treatment in a cohort of children diagnosed with ALL.

Major comments

  1. Along the manuscript the authors mentioned that they take into account the gender of the patients to perform the analysis. I would like to clarify that gender refers to the socially constructed roles, behaviours, expressions and identities of girls, women, boys, men, and gender diverse people. On the other hand, sex refers to a set of biological attributes in humans and animals. It is primarily associated with physical and physiological features including chromosomes, gene expression, hormone levels and function, and reproductive/sexual anatomy and it is usually categorized as female or male. The authors should clarify if they mean gender or sex along the manuscript.

Minor comments

  1. There are many acronyms without definition along the manuscript. (PTWG, line 88, LMWH, line 103, SCMI, RR…). Please, define them the first time they appear in the text.
  2. Figure 1 is a figure + a table. The table could be removed because its content is explained in the figure. Also, the legend of figure 1 should include the abbreviations.
  3. The lines 126-128 correspond to the legend of the table 2?
  4. Figure 4 and 4B. It should be only one figure, with one legend, not two different legends
  5. Figure 5 and 5B. It should be only one figure, with one legend, not two different legends
  6. The conclusions should be after the discussion, and not the methods section.

Author Response

Major comments

  1. Along the manuscript the authors mentioned that they take into account the gender of the patients to perform the analysis. I would like to clarify that gender refers to the socially constructed roles, behaviours, expressions and identities of girls, women, boys, men, and gender diverse people. On the other hand, sex refers to a set of biological attributes in humans and animals. It is primarily associated with physical and physiological features including chromosomes, gene expression, hormone levels and function, and reproductive/sexual anatomy and it is usually categorized as female or male. The authors should clarify if they mean gender or sex along the manuscript.

Response:

The term was corrected throughout the manuscript, as you have rightly commented (lines 83, 130, 132, 386).

 Minor comments

Comment 1: There are many acronyms without definition along the manuscript. (PTWG, line 88, LMWH, line 103, SCMI, RR…). Please, define them the first time they appear in the text.

Response:

Acronyms were defined according to your comment. (lines 88-89,111-112,126-127,133-134).

   Comment 2: Figure 1 is a figure + a table. The table could be removed because its content is explained in the figure. Also, the legend of figure 1 should include the abbreviations.

Response:

Table 1 has been removed, and abbreviations were added to the legend of figure 1 (lines 105-106).

Comment 3: The lines 126-128 correspond to the legend of the table 2?

Response:

Yes, I have adapted the font size (9) and add a spaceline after the abbreviations. (line 139)

Comment 4: Figure 4 and 4B. It should be only one figure, with one legend, not two different legends

Response:

Figures 4a-b were united with only one legend (lines 178-185).

Comment 5: Figure 5 and 5B. It should be only one figure, with one legend, not two different legends

Response:

Figures 5a-b were united with only one legend (lines 340-352).

Comment 6: The conclusions should be after the discussion, and not the methods section

Response:

The conclusion section has been moved, and now follows the discussion section (lines 312-323).

Reviewer 3 Report

Reviewer’s comments

Title: Venous thromboembolism and its risk factors in children with acute lymphoblastic leukemia in Israel: a population-based study

The authors determined that severe hypertriglyceridemia during therapy was risk factor for VTE in 388 ALL patients, although the underlying mechanism is not clear. This finding is quite interesting and should be validated in different patients’ cohort.

(Major concerns)

1.The majority of VTE is DVT. The reviewer wonders how these DVT were diagnosed. Is it possible to miss the diagnosis of DVT in certain proportion of patients in this study?

2.It is quite interesting that VTE is associated with severe hypertriglyceridemia during treatment, which might be related to the activity of L-asparaginase. It is also well known that older age (older than 10 years) is risk factor for VTE as the authors identified in this study. Do you see the association of hypertriglyceridemia and patients’ age?

Author Response

Comment 1: The majority of VTE is DVT. The reviewer wonders how these DVT were diagnosed. Is it possible to miss the diagnosis of DVT in certain proportion of patients in this study?

Response:

Thank you very much for your important comment. During our study design we have decided to focus only on clinically significant and symptomatic DVT. Accordingly, we captured and analyzed only symptomatic and significant DVT (≥ grade 2B PTWG) when systemic anticoagulation is indicated. Truly, asymptomatic DVT were missed by design but we believe it is unlikely that we have missed symptomatic events since all patients were systematically clinically monitored during therapy. Each DVT in symptomatic patients (pain, swelling of a limb) was confirmed by doppler sonography or via an echocardiogram done for all children during induction and delayed intensification periods. 

Comment 2: It is quite interesting that VTE is associated with severe hypertriglyceridemia during treatment, which might be related to the activity of L-asparaginase. It is also well known that older age (older than 10 years) is risk factor for VTE as the authors identified in this study. Do you see the association of hypertriglyceridemia and patients’ age?

Response:

Hypertriglyceridemia was not found to be associated with older age. We indeed as you wisely suggested, looked comprehensively for variables that may anticipate severe hypertriglyceridemia during therapy but neither age nor BMI% or triglyceride levels at leukemia diagnosis were identified as risk factors for hypertriglyceridemia during therapy ) Pearson correlation coefficient for age = 0.05 =ns). According to your comment we add these finding to the manuscript (line 176-177) and we also added this figure: (Figure S2)-
